# Obsessive-Compulsive Disorder with Psychotic Features: Is It a Clinical Entity?

**DOI:** 10.3390/healthcare10101910

**Published:** 2022-09-29

**Authors:** Yasushi Okamura, Yuki Murahashi, Yuna Umeda, Toshihiro Misumi, Takeshi Asami, Masanari Itokawa, Hirohiko Harima, Masafumi Mizuno, Hisato Matsunaga, Akitoyo Hishimoto

**Affiliations:** 1Department of Psychiatry, Tokyo Metropolitan Matsuzawa Hospital, 2-1-1 Kamikitazawa, Setagaya-ku, Tokyo 156-0057, Japan; 2Department of Psychiatry, Yokohama City University Graduate School of Medicine, 3–9 Fukuura, Kanazawa-ku, Yokohama 236-0004, Japan; 3Department of Biostatistics, Yokohama City University School of Medicine, 3–9 Fukuura, Kanazawa-ku, Yokohama 236-0004, Japan; 4Schizophrenia Research Project, Tokyo Metropolitan Institute of Medical Science, 2-1-6 Kamikitazawa, Setagaya-ku, Tokyo 156-8506, Japan; 5Department of Neuropsychiatry, Hyogo Medical University, 1-1 Mukogawacho, Nishinomiya 663-8501, Japan

**Keywords:** obsessive-compulsive disorder, psychotic disorder, schizophrenia, comorbidity, insight, functioning

## Abstract

(1) Background: Even though the comorbidity of obsessive-compulsive disorder (OCD) and a psychotic disorder (PD), such as schizophrenia, is being increasingly recognized, the impact of this comorbidity on the clinical presentation, including insight into obsessive-compulsive symptoms and the functioning of OCD, remains unclear. (2) Methods: To investigate clinical differences between OCD patients with and without PD, 86 Japanese outpatients who met the DSM-IV-TR criteria for OCD were recruited and divided into two groups: 28 OCD patients with PD, and 58 OCD patients without PD. The two groups were cross-sectionally compared in terms of their sociodemographic profiles and clinical characteristics, including the DSM-IV-TR insight specifier and the Global Assessment of Functioning (GAF). (3) Results: The results showed that OCD patients with PD scored lower on both the insight and GAF assessments. (4) Conclusions: The present study suggests that comorbid PD in OCD is a clinical entity.

## 1. Introduction

### 1.1. Background

The relationships between obsessive-compulsive disorder (OCD) and various psychotic disorders (PDs), such as schizophrenia, have long been noted [1,2,3,4,5,6]. OCD, schizophrenia, and their comorbidity interact with each other in a manner that may affect prognosis and treatment [7,8,9,10]. The comorbidity of OCD and schizophrenia is currently being gradually recognized [11], and a common biological basis may underlie the higher-than-expected comorbidity rate [12,13].

Most of previous studies have focused on patients with schizophrenia, with or without obsessive-compulsive symptoms or OCD [14], but not vice versa. It is clear from the assessments by a number of investigators over the last two decades that a subgroup of patients with schizophrenia holds co-occurring obsessions and compulsions, while early studies on psychotic symptoms in patients with primary OCD did not use standardized diagnostic criteria [15]. Prior to the DSM-III-R [16], the diagnosis of OCD was ruled out by the presence of schizophrenia, and obsessive-compulsive symptoms in patients with schizophrenia were interpreted as symptoms of schizophrenia, as is still the case in ICD-10 [13,17]. Therefore, research on the comorbidity and relationship between the two disorders has mainly emphasized broad variations in the psychopathological aspects of schizophrenia. DSM-5 [18] noted that the prevalence of OCD was higher in patients with schizophrenia than in that in a general population. Achim et al. (2011) showed that 12% of patients with schizophrenia had OCD. While obsessive-compulsive symptoms in schizophrenia have been proposed as a defense against psychotic deterioration [19] and, thus, are a predictor of a positive prognosis [20,21], many studies reported that the comorbidity of OCD among patients with schizophrenia had a negative impact on their prognosis [8,22,23,24]. Claims have been made [25,26,27,28,29] that the comorbidity of schizophrenia and OCD is a subtype of schizophrenia, with specific clinical features, but not without disagreement [30]. 

Less attention has been paid to OCD patients who are with or without PD. Comorbid PDs are frequently indicated as exclusion criteria in the majority of clinical research on OCD. Hence, limited information is currently available on the clinical characteristics of OCD patients with schizophrenia [13,31]. Numerous studies on psychotic traits comorbid with OCD have focused on schizotypal personality traits [32,33,34,35]. Epidemiological studies showed that 1–12.5% of patients with previously diagnosed OCD developed PD [36], and about one out of ten (12%) of patients with OCD met the diagnostic criteria for schizophrenia [25]. Recent meta-analyses found that individuals with OCD are more likely to have psychosis than the general population [37], and recent large-scale studies found that OCD increased the risk of developing schizophrenia after the onset of OCD [38,39,40,41], which is in contrast to the findings of previous studies showing that OCD was not associated with an increased risk of schizophrenia [42,43]. OCD with schizophrenia was more common in men [7], was associated with a lower score on the Global Assessment of Functioning (GAF) [9], had a more deteriorative course [7], was more resistant to conventional OCD treatments [44], and was susceptible to the exacerbation of psychotic symptoms when an anti-OCD agent was administered [10].

As used in Matsunaga et al.’s (2002) study, the GAF is internationally well known and widely used for scoring the severity of illness in psychiatry [45], and is recommended for routine clinical use [46]. The GAF summarizes the clinician’s view of the patient’s current degree of impairment in terms of psychosocial and occupational or educational function. Despite this, the GAF has been significantly less used in OCD studies. 

Regarding insight into the obsessive-compulsive symptoms of pivotal clinical importance [42], Matsunaga et al. (2002) reported that a large percentage of patients with OCD had poor insight [9], and this may affect treatment and prognosis [47]. However, many studies discussing insight in patients with OCD excluded a psychotic comorbidity [35,48,49,50,51]. Studies on insight in patients with OCD with PD, especially schizophrenia, are rare [9,52]. It is still debatable whether comorbid PD is associated with a better [13] or worse [7,9,30] outcome of OCD. 

A specifier of OCD with poor insight was first introduced in DSM-IV-TR [53]. The DSM-5 provides 3 insight specifiers: (1) good to fair, (2) poor, and (3) absent/delusional beliefs. Similarly, the ICD-11 has recently employed simpler dichotomous insight specifiers: (1) good to fair insight or (2) poor to absent insight [47]. As far as we know, there is one previous study on clinical characteristics and insight in patients with OCD using dichotomous insight specifiers [49]; this study, however, evaluated only OCD patients without PDs.

### 1.2. Objectives

The present study compared clinical characteristics between OCD patients with and without PD to identify what would differentiate the two groups, with particular focus on insight into OCD and the Global Assessment of Functioning. Our hypotheses were that OCD patients with PD would show poorer insight and lower GAF compared with OCD patients without PD.

## 2. Methods

### 2.1. Participants

Between April 2015 and April 2022, 86 outpatients at the Department of Psychiatry at Tokyo Metropolitan Matsuzawa Hospital were enrolled after submitting their written informed consent to participate in this study. Our hospital, the largest psychiatric center in Tokyo, receives approximately 7600 new patients annually, including 45 new OCD patients. After conducting surveys of outpatient medical records, all patients had been diagnosed with OCD (based on the DSM-IV-TR criteria) independently by an experienced psychiatrist (Y.O.) with more than five years of experience in the treatment of OCD, who was different from the attending psychiatrists. Inclusion criteria were patients with OCD based on the DSM-IV-TR criteria. No specific exclusion criteria were established. All patients were assessed using the Mini International Neuropsychiatric Interview, Japanese version 5.0.0 2003 (MINI), administered by one of the authors (Y.O.) or the attending psychiatrist. The MINI is a reliable and valid structured interview that may be administered by clinicians or trained non-clinicians to screen for 17 of the most common mental disorders listed in the ICD-10 and DSM-IV-TR [54,55,56,57,58]. The interview confirmed the diagnosis of OCD in all patients for the previous month, except for six who clearly had a history of OCD symptoms and irrational feelings about their obsessive-compulsive symptoms, but who were free from OCD symptoms in the month prior to the interview. Comorbidities not covered by the MINI were diagnosed according to the ICD-10. We also ensured that in OCD patients with PD, psychotic symptoms did not consist only of the delusional nature of insight into their obsessive-compulsive symptoms, but included other symptoms related to PD, including delusions other than those only related to poor insight, hallucinations, thought disorders, and negative symptoms. A total of 86 patients with OCD were divided into 2 groups: 28 patients with comorbid PD (26 with schizophrenia, one with schizotypal disorder, and one with schizoaffective disorder, depressive type, according to ICD-10) and 58 patients without PD.

### 2.2. Ethical Considerations

All of the procedures in the present study complied with the ethical standards of the relevant national and institutional committees on human experimentation and were conducted according to the guidelines of Declaration of Helsinki. The present study was approved by the Ethics Committee of Tokyo Metropolitan Matsuzawa Hospital. Detailed explanations of the study procedures were provided to each participant prior to their informed consent.

### 2.3. Clinical Evaluation

A detailed interview was conducted by one of the authors (Y.O.) covering the information on the patients’ demographic profiles, clinical features, social background factors, and family and medical histories, as well as the clinical course and characteristics of their OCD. Insight levels were dichotomously assessed using the DSM-IV-TR insight specifier, which defines poor or absent/delusional insight as an individual’s lack of awareness that his or her obsessions and compulsions are irrational during most episodes of OCD [53]. Patients other than those with poor to absent/delusional insight were assigned to the group characterized as having good to fair insight. If a patient had multiple obsessive-compulsive symptoms and no insight into at least one of these symptoms, the patient was defined as having poor to absent/delusional insight. Then, 20 out of 86 participants were randomly selected from the patients attending in January and February 2022 to examine the inter-rater reliability of the assessment of insight based on a joint interview by two of the authors (Y.O. and Y.M.), and the kappa coefficient [59] was 0.88.

Global severity and the prevalence of obsessive and compulsive symptoms was evaluated using the Yale–Brown Obsessive Compulsive Scale (Y-BOCS) [60]; general functioning was assessed using the Global Assessment of Functioning (GAF) [16,61,62]; disease severity was evaluated using the Clinical Global Impressions and Severity of Illness (CGI-S) [63]; and the severity of depression was rated using the Japanese version of the GRID-Hamilton Rating Scale for Depression (GRID-HAMD) [64,65], a structured interview incorporating the Hamilton Rating Scale for Depression (HAMD) [66], the international gold standard for assessing the severity of depression with high inter-rater reliability.

### 2.4. Statistical Analysis

Continuous variables were summarized with means (standard deviation, SD) regarding OCD with and without PD, and compared using a *t*-test between groups. Categorical variables were presented as frequencies and percentages by groups, and compared using the chi-square test or Fisher’s exact test. The significance level was set at *p* < 0.05. A multivariate logistic regression analysis was used to extract the most influential clinical variables distinguishing OCD, with and without a psychotic comorbidity. The independent variables used were all seven variables that were significant in the bivariate analysis— the five clinical factors reported in previous studies as significant for distinguishing the two groups: gender, marital status, age at OCD onset, GAF, and CGI-S, plus the two new items significantly differing between the two groups in the present study, namely, insight and an involuntary initial visit to a healthcare provider for a consultation regarding obsessive-compulsive symptoms. Statistical analyses were performed using IBM SPSS Statistics (Version 23; IBM Corporation, Armonk, NY, USA, 1989, 2015) (SPSS Inc., Chicago, IL, USA). 

## 3. Results

### 3.1. Sociodemographic and Clinical Characteristics

No significant differences were observed in age, housemates, education, employment, or a medical or family history of a psychiatric disease, or comorbidity of psychiatric disorders other than PD between the OCD cases with and without PD (Table 1). The gender ratio was different between the two groups: men were overrepresented in OCD patients with PD. None of the OCD patients were married among OCD patients with PD, while one quartier of the OCD patients without PD were married.

### 3.2. OCD-Related Aspects

A total of 41 out of the 86 patients examined (47.7%) had poor to absent/delusional insight. Patients in the OCD with PD group were significantly more likely than those without PD to have poor to absent/delusional insight (23/28 vs. 18/58, *p* < 0.001) (Table 2).

The mean age of onset of OCD was lower in OCD patients with PD. The mean (SD) duration of untreated OCD was not different between the two groups. OCD patients with PD were more likely than those without PD to initially make an involuntary visit to a healthcare provider for a consultation regarding obsessive-compulsive symptoms (Table 2).

OCD patients with and without PD were not different in terms of the mean Y-BOCS scores and the types of OCD symptoms, as well as the mean (SD) GRID-HAMD score (Table 2).

The GAF score was significantly lower, and the CGI-S was significantly higher in the OCD with PD group than in the OCD without PD group (Table 2).

### 3.3. Temporal Course of OCD and Schizophrenia

Among the 26 OCD patients with schizophrenia, the onset of OCD preceded schizophrenia in a majority of cases (22 patients); the onset of the disorders was simultaneous in two patients, and the onset of schizophrenia preceded OCD in two patients. In the 22 patients in whom the onset of OCD preceded schizophrenia, the mean (SD) age of OCD onset was 15.0 (5.5) years. The mean (SD) delay of schizophrenia onset after OCD was 9.7 (7.5) years, and the mean (SD) onset age of schizophrenia was 24.7 (10.4) years.

### 3.4. Multivariate Analysis

The multivariate logistic regression analysis was performed to identify the clinical variables that best distinguish OCD with PD from OCD without PD (Table 3). Two items significantly predicted group membership: poor to absent/delusional insight (odds ratio: 0.065; *p* < 0.001) and lower GAF (odds ratio: 0.927; *p* = 0.012).

## 4. Discussion

The main results of this study were that OCD patients with PD scored lower in insight and GAF evaluations compared with OCD patients without PD.

### 4.1. Insight

In the present study, 41 patients (47.7%) were considered to have poor to absent/delusional insight, which is consistent with previous findings. Matsunaga et al. (2002) reported that a large percentage of patients with OCD had poor insight. Although a lack of insight is generally assumed to be rare in patients with OCD [7], recent studies demonstrated that the frequency of poor insight was 15–31% in patients with OCD without PD [48,49,50,51] and 9–36% in patients with OCD with PD [9,52]. The present results showed a slightly higher prevalence of OCD patients with poor insight, which may be attributed to the higher rate of psychotic comorbidity (32.6%) than in previous studies (20% [9] and 1.7% [52]). The prevalence rate of poor insight in patients with OCD may be influenced by the degree of treatment for OCD at the time of interview, since previous studies have reported that insight can be improved after treatment for OCD [9].

The results of the present study are consistent with previous findings showing that OCD with schizophrenia was correlated more strongly with poor insight than OCD without schizophrenia [9,30,67]. As for the mechanism of poor insight in OCD with schizophrenia, it is possible that the schizophrenic thought disorders may contribute to the poor insight into OCD. Further study considering the degree and effect of thought disorders due to schizophrenia on insight into OCD are required.

The assessment of insight or comorbid schizophrenia in OCD is important because it is relevant to treatment planning. Some previous studies reported that OCD with poor insight was closely associated with a poorer response to medication [50], and some reported that OCD patients with schizophrenia were often resistant to typical OCD treatments [44]. As for pharmacological treatment, for example, in the American Psychiatric Association practice guideline for the treatment of patients with OCD, the pharmacological treatment of OCD with poor insight or with comorbid schizophrenia has not been established, and psychiatrists must rely on clinical judgment in formulating a treatment plan, since no large, controlled trials have yet been conducted [68]. Some reports stated that the use of an augmenting atypical antipsychotic was effective in patients with poor insight and an early age of OCD onset [69], and olanzapine monotherapy has been beneficial for patients with co-occurring schizophrenia in two case series, while second-generation antipsychotics were reported to exacerbate obsessive-compulsive symptoms [68]. Therefore, since there is no consensus on pharmacological treatment of OCD patients with poor insight or with the comorbidity of schizophrenia, those groups of patients may be resistant to conventional OCD treatment, and further research is needed to improve pharmacological therapeutic strategies.

### 4.2. Sociodemographic Profiles and Clinical Characteristics

In the present study, the OCD group with PD was significantly more likely than the OCD group without PD to exhibit the following clinical characteristics: unmarried, male, with poor prognostic factors, including a lower GAF and a higher CGI-S score in the univariate analysis, which is consistent with previous findings [7,9,30]. Among these factors, the multivariate logistic regression analysis revealed that the GAF score was a significant predictor of the OCD group with PD, suggesting the negative clinical impact of the comorbidity of PD and OCD. Since GAF measures the degree of mental illness by rating psychological, social, and occupational functioning [45], further research is needed to evaluate this issue using more specific rating scales that evaluate each function separately. 

In the present study, the mean untreated duration of OCD was 7.1 years, which closely corresponded to the 7 years reported by previous studies [70]; however, the period may be as long as 17 years [71]. The prevalence of OCD in the general population is reportedly as high as 1.1–1.8% [18] or 2.5% [25]. Many patients with OCD hesitate to seek medical care and exhibit a low rate of hospital visits. Comorbidities may complicate the disease and make it enduring, and a chronic course is one of the poorest prognostic factors [72]. Therefore, early interventions, such as sharing knowledge about OCD and the importance of early consultation as part of medical care, with the general public and family physicians and enabling early access to treatment by OCD experts, is important for the prevention of chronicity and severity in OCD.

### 4.3. Temporal Course of OCD and Schizophrenia

There are four possibilities regarding the pathogenesis of the comorbidity of OCD and schizophrenia. First, the two disorders may coexist by chance, without affecting each other. Second, OCD may develop first as a prodromal symptom of schizophrenia, or schizophrenia with OCD may represent a subtype of schizophrenia [28]. Previous studies on OCD in individuals at ultra-high risk of psychosis reported various conclusions [73,74]. Third, coexistent OCD and schizophrenia may have a mutually exacerbating or amplifying effect [29,75], and persistent OCD may predispose a patient to an increased risk of developing schizophrenia [40,41,73]. OCD may also become schizophrenia [76] or another form of psychosis [42]. Finally, patients with the comorbidity of OCD and schizophrenia may have common risk factors or a neurobiological basis [12,77,78]. In the present study, the onset of OCD preceded that of schizophrenia in most of the cases with PD, which is in line with previous findings [39,79,80]. In comorbidity cases, average ages at the onsets of OCD and schizophrenia were 15.0 years (slightly lower than the DSM-5 figure of 19.5 years, which was similar to those reported in previous studies [7,13]) and 24.7 years, respectively. Among the above four possibilities, the second, third, and fourth possibilities are the most likely, and the coexistence of the two disorders suggests more than a chance occurrence, because OCD with PD may be distinct from OCD without PD, based on poor to absent/delusional insight and the lower GAF score found in the present study. In addition, previous reports may configure such OCD with PD groups characterized by male predominance, younger age at OCD onset, and poor prognosis. To explore these possibilities, increasing the sample size and conducting cluster analysis or latent class analysis in a heterogeneous group of OCD patients with various comorbidities may also be useful in elucidating the pathophysiology of OCD. Such studies would provide a more detailed picture of the relationship between the two disorders, or of the cases of comorbidity of OCD and schizophrenia, especially in patients characterized by poor insight, low functioning, male predominance, younger age at OCD onset, and poor prognosis. 

### 4.4. Strengths and Limitations

The present study has three strengths. First, it is one of the few studies to focus on the clinical characteristics of OCD with and without PD, in contrast to the large number of previous studies on schizophrenia with and without OCD. Recognizing the difference between OCD patients with and without PD may improve current treatments. For example, clinicians may provide adequate psychoeducation to patients with a comorbidity of OCD and psychosis in order to improve their insight, or consider a sufficient number of treatment options, such as the addition of antipsychotics if patients are resistant to conventional OCD treatment, or introduce the use of measures to prevent the development of schizophrenia. 

Second, the present study contains rather common and practical information on clinical judgement in daily psychiatric settings. Psychiatrists need to make treatment plans for patients with OCD with various comorbidities, including PD, and to seek multidimensional information about the patients based on the results of the GAF; with these processes, we believe, better treatment of the patients with OCD can be achieved.

Third, the present study was conducted at a general psychiatric center in Japan, with no outpatient clinic specializing in OCD treatment. In Japan, there are not sufficient OCD specialists, and many psychiatric institutions do not have outpatient clinics specializing in OCD. Patients with OCD hesitate to disclose their obsessive-compulsive symptoms, resulting in a long period of time before they receive specialized treatment. Furthermore, clinicians may overlook obsessive-compulsive symptoms because OCD patients with poor insight may present to general psychiatric outpatient clinics for other prominent psychiatric symptoms, such as depression and psychomotor agitation. While this study was conducted in a more general psychiatric institution with no outpatient clinic specializing in OCD, it is important to note that our hospital is a medical institution that treats many patients with schizophrenia, particularly the most severe forms of the disorder.

The present study is not without limitations. It contained a selection bias. The study center is the largest general psychiatric center in Tokyo, but has no specialized OCD treatment clinic. We included one patient with schizotypal disorder and one with schizoaffective disorder under rubric of PD. This may have biased the results. However, we obtained virtually the same results (Appendix A) when we repeated the analysis using 26 patients with schizophrenia and 58 OCD patients without PD (thus excluding the above 2 patients). Our OCD patients with PD included four patients with OCD onset that did not precede the onset of PD. This may have biased the results. However, we obtained virtually the same results (Appendix A) when we repeated the analysis using 24 patients whose OCD onset preceded that of PD and 58 OCD patients without PD (thus excluding the above 4 patients). Furthermore, the present study was exploratory; hence, the results of the multivariate logistic regression were not conclusive. Moreover, this study did not use a more detailed OCD insight scale. Notwithstanding these limitations, we believe this study provides clinically essential results, with practical applications.

## 5. Conclusions

The present study demonstrated that patients with a comorbidity of OCD and psychotic disorders were more likely to have poor insight and a lower GAF score than patients with OCD without psychotic disorders, and suggested that a psychotic disorder is one of the important clinical factors in assessing patients with OCD. The present results indicate that patients with this comorbidity may have a clinically different phenotype from that of patients with OCD without psychotic disorders. Future studies that enroll a larger cohort and employ more quantitative and qualitative assessment scales or multicenter research, including other institutions specializing in OCD clinics, are needed to generalize the results.

## Figures and Tables

**Table 1 healthcare-10-01910-t001:** Comparison between OCD with psychotic disorder (*n* = 28) and OCD without psychotic disorder (*n* = 58): sociodemographic profiles and clinical characteristics.

	OCD with PD (*n* = 28) (%)	OCD without PD (*n* = 58) (%)	*p*
Gender			
Men	20 (71.4)	26 (44.8)	0.02
Women	8 (28.6)	32 (55.2)	
Age, mean (SD)	29.7 (10.7)	31.2 (11.9)	0.579
Marital status			
Married	0 (0.0)	14 (24.1)	0.004
Unmarried	28 (100.0)	44 (75.9)	
Housemates			
Living alone	4 (14.3)	11 (19.0)	0.765
Cohabiting or living in an institution	24 (85.7)	47 (81.0)	
Educational level			
High school or higher, including current students	20 (71.4)	51 (87.9)	0.059
Junior high school, including current students	8 (28.6)	7 (12.1)	
Present employment status			
Employed, housewife, or current student	5 (17.9)	18 (31.0)	0.298
Unemployed or on a leave of absence from duty or school	23 (82.1)	40 (69.0)	
Physical comorbidity			
Yes	10 (35.7)	18 (31.0)	0.584
No	17 (60.7)	40 (69.0)	
Family history of psychiatric illness			
Yes	18 (64.3)	29 (50.0)	0.212
No	10 (35.7)	29 (50.0)	
Psychiatric comorbidity			
Mood disorder	13 (46.4)	27 (46.5)	0.991
Anxiety disorder	10 (35.7)	16 (27.6)	0.462
Autism spectrum disorder	3 (10.7)	14 (24.1)	0.247
Self-harm attempt			
Yes	12 (42.9)	18 (31.0)	0.281
No	16 (57.1)	40 (69.0)	

The *t*-test or chi-square test were used to compare the groups, and Fisher’s exact test was used if there were cells with expected frequencies of five or less.

**Table 2 healthcare-10-01910-t002:** Comparison between OCD with psychotic disorder (*n* = 28) and OCD without psychotic disorder (*n* = 58): clinical features and measures.

Variables	OCD with PD (*n* = 28)	OCD without PD (*n* = 58)	*p*
Insight into obsessive compulsive symptoms, *n* (%)			
Poor to absent/delusional	23 (82.1)	18 (31.0)	<0.001
Good to fair	5 (17.9)	40 (69.0)	
Age at OCD onset, mean (SD)	16.3 (6.2)	19.9 (9.4)	0.033
Duration of untreated OCD (yr), mean (SD)	7.4 (7.4)	5.6 (7.4)	0.295
Duration of OCD (yr), mean (SD)	14.0 (9.8)	11.3 (9.9)	0.232
First consultation with a health care provider for OCD			
Voluntary	9 (32.1)	35 (60.3)	0.014
Involuntary	19 (67.9)	23 (39.7)	
GRID-HAMD, mean (SD)	16.9 (8.6)	16.8 (8.9)	0.961
Y-BOCS, mean (SD)	27.5 (9.7)	26.2 (9.0)	0.555
Types of obsession, *n* (%)			
Aggression	11 (39.3)	24 (41.4)	0.853
Contamination	22 (78.6)	39 (67.2)	0.278
Sexual	1 (3.6)	2 (3.4)	1.000
Hoarding	2 (7.1)	11 (19.0)	0.207
Religious	1 (3.6)	1 (1.7)	0.548
Symmetry/exactness	4 (14.3)	16 (27.6)	0.276
Somatic	4 (14.3)	11 (19.0)	0.765
Miscellaneous	9 (32.1)	21 (36.2)	0.711
Types of compulsion, *n* (%)			
Cleaning/washing	22 (78.6)	38 (65.5)	0.217
Checking	12 (42.9)	34 (58.6)	0.170
Repeating	11 (39.3)	24 (41.4)	0.615
Counting	1 (3.6)	5 (8.6)	0.659
Ordering/arranging	2 (7.1)	6 (10.3)	1.000
Hoarding/collecting	2 (7.1)	7 (12.1)	0.712
Miscellaneous	5 (17.9)	8 (13.8)	0.749
GAF, mean (SD)	22.7 (13.8)	38.0 (15.9)	<0.001
CGI-S, mean (SD)	6.7 (0.48)	6.1 (1.1)	<0.001

GAF = Global Assessment of Functioning; CGI-S = Clinical Global Impressions of Severity scale; GRID-HAMD = GRID-Hamilton Rating Scale for Depression; Y-BOCS = Yale–Brown Obsessive-Compulsive Scale. The *t*-test or chi-square test was used to compare the groups, and Fisher’s exact test was used if there were cells with expected frequencies of five or less.

**Table 3 healthcare-10-01910-t003:** Multivariate logistic regression analysis of factors of OCD with psychotic disorder (*n* = 28) and OCD without psychotic disorder (*n* = 58): sociodemographic profiles, clinical features, and measures.

Variables	Level	Odds Ratio	95%CI	*p*
Insight into obsessive compulsive symptoms	Poor to absent/delusional vs. good to fair	0.065	0.013–0.318	0.001
Gender	Male vs. female	0.443	0.123–1.604	0.215
GAF		0.927	0.874–0.983	0.012
CGI-S		0.895	0.262–3.053	0.859
Age at OCD onset		0.969	0.881–1.065	0.513
First consultation for OCD	Voluntary vs. involuntary	0.513	0.113–2.328	0.387
Marital status	Unmarried vs. married	0.000	0.000	0.998

CI = confidence interval.

## Data Availability

All data were generated at the Tokyo Metropolitan Matsuzawa Hospital, Japan. The derived data supporting the findings of this study are available from the corresponding author on request.

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
