# Peer review of "Obsessive-Compulsive Disorder with Psychotic Features: Is It a Clinical Entity?"

_healthcare, 2022, doi:10.3390/healthcare10101910_

Round 1
Reviewer 1 Report
a readable and engaging paper. The perspective of the authors (OCD and PD) instead of the conventional PD and OCD makes for an exciting read.
This is an interesting paper in which they set out to establish a distinction of two similar syndromes when experiencing added psychotic symptomatology.
I think is a relevant topic that brings to the fore the cyclical nature of our discipline. In a way, this paper revisits a convergence (between a group of words - i.e., obsession, zwangsvorstellung, impulsions, etc.-, and some enduring behaviors) that took place in the last quarter of the XIXth century (Berrios 1985 and 1995). Besides, it alerts clinicians of the need to review in extension the symptomatology of patients with complex psychopathological presentations.
It is an interesting perspective because it explores the phenomenon of comorbidity from the side of the “Neuroses” instead of the all-encompassing side of the “Psychoses.”
No specific improvements could the authors consider regarding the methodologyspecific improvements come to mind.
The conclusions are fairly consistent with the weight of the evidence and arguments presented.
The references are appropriate and current.
Berrios, G. E. Obsessional disorders during the nineteenth century: terminological and classificatory issues. In: The Anatomy of Madness. Volume I. People and Ideas. Ed: Bynum, W.F. , Porter R, Shepherd M. Tavistock Publications. London and New York, 1985.
Berrios, G.E. Obsessive-Compulsive Disorder. In: A History of Clinical Psychiatry. Ed: Berrios G.E. and Porter R. The Athlone Press, London, 1995.
No additional comments on the tables and figures.
Author Response
Thank you very much for your peer review. I would like to provide a point-by-point response to your comments:
Berrios, G. E. Obsessional disorders during the nineteenth century: terminological and classificatory issues. In: The Anatomy of Madness. Volume I. People and Ideas. Ed: Bynum, W.F. , Porter R, Shepherd M. Tavistock Publications. London and New York, 1985.
Berrios, G.E. Obsessive-Compulsive Disorder. In: A History of Clinical Psychiatry. Ed: Berrios G.E. and Porter R. The Athlone Press, London, 1995.
: We have newly added these appropriate references.
Reviewer 2 Report
I have completed my evaluation of a manuscript titled “Obsessive-compulsive disorder with psychotic features: Is it a clinical entity?”. The motivation for this study is warranted and an important area of research. These findings can make an important contribution to the literature.
1. Introduction
The introduction is well written, but it needs improvement. The authors provide a broad overview of research on obsessive-compulsive disorder and psychotic disorder. This is somewhat satisfactory; however, the authors do not clearly identify any gaps in the literature that they hope to fill.
Additionally, in paragraph 1.2. Objectives, there was a lack of specific hypotheses.
2. Methods
2.1. Participants
Participants were described insufficiently. How participants were recruited? What were the inclusion and exclusion criteria? Please describe. How many patients are under supervision in this hospital, out clinic? How many patients does the clinic provide services for?
2.3. Clinical evaluation
The author noticed: “participants were randomly selected” (line 133). How? Please describe.
2.4. Statistical analysis
In statistical analysis missed the description of analysis distribution and description of the way of descriptive analysis.
Additionally, the sentence (line 146-147) “The 86 patients with OCD were divided into 2 groups: 28 patients with comorbid PD (26 with schizophrenia, one with schizotypal disorder, and one with schizo-affective disorder according to ICD-10) and 58 patients without PD.“ do not belong in the statistical method. It would be better to take out this sentence from this paragraph, and put it into paragraph 2.1. Participants or 2.3. Clinical evaluation.
3. Results
Line 191-197 is confusing. This belongs to the statistical method – not to results. “The multivariate logistic regression analysis was performed to identify the clinical variables that best distinguish OCD with PD from OCD without PD (Table 3). The independent variables used were the five clinical factors reported in previous studies as significant for distinguishing the two groups: gender, marital status, age at OCD onset, GAF, and CGI-S, plus the two new items significantly differing between the two groups in the present study, namely, insight and an involuntary initial visit to a healthcare provider for a consultation regarding obsessive-compulsive symptoms”. So, please think about reformulation, and part of this paragraph should move into the statistical method.
Tables
In the footer of all tables need to explain which statistical test the authors used.
4. Discussion
At the beginning of the discussion, I miss the "red line". It would be improved if you start to describe the main results of your study.
Also, to me, it does not make much sense to divide the discussion into a few paragraphs. This also applies to the introduction, too. The conclusion needs to be independent of discussion.
Author Response
Thank you very much for your peer review. I would like to provide a point-by-point response to your comments:
- Introduction
The introduction is well written, but it needs improvement. The authors provide a broad overview of research on obsessive-compulsive disorder and psychotic disorder. This is somewhat satisfactory; however, the authors do not clearly identify any gaps in the literature that they hope to fill.
As a result of your thoughtful guidance, in order to identify any gaps in the literature, we have cited a section of the following textbook as a reference and have added the following sentence into the second paragraph of 1.1. Background:
Eisen, J.E.Y., A. G.; Mancebo, M. C.; Pinto, A.; Rasmussen, S. A. Phenomenology of Obsessive-Compulsive Disorder; American Psychiatric Publishing: Arlington, 2010
It is clear that a subgroup of patients with schizophrenia has co-occurring obsessions and compulsions from the assessment by a number of investigators over the last two decades, while early studies on psychotic symptoms in patients of primary OCD used standardized diagnostic criteria [15].
Additionally, in paragraph 1.2. Objectives, there was a lack of specific hypotheses.
: We have newly added the following sentence in paragraph 1.2. Objectives: Our hypotheses were that OCD patients with PD would show poorer insight and lower GAF compared with OCD patients without PD.
- Methods
2.1. Participants
Participants were described insufficiently. How participants were recruited? What were the inclusion and exclusion criteria? Please describe. How many patients are under supervision in this hospital, out clinic? How many patients does the clinic provide services for?
: We recruited participants by conducting surveys of outpatient medical records and the inclusion criteria were patients with OCD based on DSM-IV-TR criteria. No specific exclusion criteria were established. We added as follows: our hospital receives approximately 7,600 new patients annually, including 45 new OCD patients.
2.3. Clinical evaluation
The author noticed: “participants were randomly selected” (line 133). How? Please describe.
: The patients were randomly selected from those attending in January and February 2022, and we have indicated this.
2.4. Statistical analysis
In statistical analysis missed the description of analysis distribution and description of the way of descriptive analysis.
: The description was revised in consultation with one of the co-authors, a statistician.
Additionally, the sentence (line 146-147) “The 86 patients with OCD were divided into 2 groups: 28 patients with comorbid PD (26 with schizophrenia, one with schizotypal disorder, and one with schizo-affective disorder according to ICD-10) and 58 patients without PD.“ do not belong in the statistical method. It would be better to take out this sentence from this paragraph, and put it into paragraph 2.1. Participants or 2.3. Clinical evaluation.
: As you pointed out, we have taken out this sentence from this paragraph, and put it into paragraph 2.1.
- Results
Line 191-197 is confusing. This belongs to the statistical method – not to results. “The multivariate logistic regression analysis was performed to identify the clinical variables that best distinguish OCD with PD from OCD without PD (Table 3). The independent variables used were the five clinical factors reported in previous studies as significant for distinguishing the two groups: gender, marital status, age at OCD onset, GAF, and CGI-S, plus the two new items significantly differing between the two groups in the present study, namely, insight and an involuntary initial visit to a healthcare provider for a consultation regarding obsessive-compulsive symptoms”. So, please think about reformulation, and part of this paragraph should move into the statistical method.
: We have moved part of this paragraph into the statistical method.
Tables
In the footer of all tables need to explain which statistical test the authors used.
: We have added explanation for which statistical test the authors used.
- Discussion
At the beginning of the discussion, I miss the "red line". It would be improved if you start to describe the main results of your study.
: At the beginning of the discussion, we have added the main results as follows: The main results of this study were that OCD patients with PD were scored poorer in insight and GAF compared with OCD patients without PD.
Also, to me, it does not make much sense to divide the discussion into a few paragraphs. This also applies to the introduction, too. The conclusion needs to be independent of discussion.
: We have made the conclusion be independent of discussion.
Reviewer 3 Report
This study examined an issue that received less attention before. This study provided evidence to differentiate between patients with and without psychotic disorders. The study design was reasonable.
I have some suggestions for the authors to improve their manuscript.
1. “All patients had been diagnosed with OCD based on the DSM-IV-TR criteria.” Please explain why the authors sued DSM-IV-TR but not DSM-5.
2. Abstract: “Results: The results showed that OCD patients with PD were scored poorer in insight and GAF.” The results were too simple.
3. “It is still debatable whether comorbid PD is associated with better (Ozdemir et al., 2003) or worse (Frias et al., 2014; Matsunaga et al., 2002; Rasmussen & Eisen, 1992) outcome of OCD.” The reference format needed to be revised.
4. Error: “No significant differences were observed in gender,…”
Author Response
Thank you very much for your peer review. I would like to provide a point-by-point response to your comments:
1. “All patients had been diagnosed with OCD based on the DSM-IV-TR criteria.” Please explain why the authors sued DSM-IV-TR but not DSM-5.
: We believe it would have been better to use DSM-5 as you mentioned. However, when the study started in 2015, the Japanese version of DSM-5 was not fully available and the hospital staff was used to DSM IVTR, so we chose to use DSM-IV-TR.
2. Abstract: “Results: The results showed that OCD patients with PD were scored poorer in insight and GAF.” The results were too simple.
: You are correct. However, the sample size of this study is small, and we would like to expand the study by increasing the number of samples and conducting complex statistical analyses such as cluster analysis in the future. We would appreciate it if you could wait for our future research to meet your expectations.
3. “It is still debatable whether comorbid PD is associated with better (Ozdemir et al., 2003) or worse (Frias et al., 2014; Matsunaga et al., 2002; Rasmussen & Eisen, 1992) outcome of OCD.” The reference format needed to be revised.
: We have revised the reference format which you ponited out.
4. Error: “No significant differences were observed in gender,…”
: We have deleted "gender" from this sentence.